# Force–Velocity Relationship in Cycling and Arm Cranking: A Comparison of Men and Women

**DOI:** 10.3390/jfmk8040151

**Published:** 2023-10-30

**Authors:** Jan Heller, Ivana Kinkorova, Pavel Vodicka, Pantelis Theodoros Nikolaidis, Stefan Balko

**Affiliations:** 1Faculty of Physical Education and Sport, Charles University, 162 52 Prague, Czech Republic; kinkorova@ftvs.cuni.cz (I.K.); pvdck@seznam.cz (P.V.); 2School of Health and Caring Sciences, University of West Attica, 122 43 Athens, Greece; pademil@hotmail.com; 3Faculty of Education, Jan Evangelista Purkyne University, 400 96 Usti nad Labem, Czech Republic; stefan.balko@ujep.cz

**Keywords:** force–velocity characteristics, males, females, cycling, arm cranking

## Abstract

This study was aimed at comparing the force–velocity relationship during cycling and arm cranking in males and females. Thirty-two male and twenty-two female healthy volunteers performed a force–velocity test on a cycle ergometer and a cranking ergometer in a randomly selected order. The theoretical values of the maximum force at zero speed (F_0_) and the maximum velocity at zero braking force (v_0_) for the lower and upper limbs were determined, and the maximum anaerobic power (Pmax) was calculated from the individual force–velocity relationship. The Pmax and F_0_ of the upper limbs related to the lower limbs correspond to 78.2 ± 14.3% and 80.1 ± 17.3% in men and 65.5 ± 12.5% and 74.5 ± 6.5% in women, respectively. The theoretical maximum velocity v_0_ of the upper limbs to the lower limbs attained 129.1 ± 29.0% in men and 127.4 ± 26.4% in women. The results of the study can serve as reference data for the force–velocity characteristics of the upper and lower limbs of male and female athletes. The results can be used both in training and rehabilitation programs, where the starting point is the objectification of possible strength deficits in various areas of the force–velocity characteristic spectrum of the muscles of the upper and lower limbs.

## 1. Introduction

Many sports and work activities place high demands on the so-called “speed-strength”, i.e., the ability to achieve and maintain a high level of strength at a high speed of movement. The force–velocity relationship was first developed for an isolated muscle [1,2] and later for a monoarticular type of movement [3]. In recent decades, the concept of force and speed has also been developed for multi-articular types of movements, especially for work on a cycle ergometer [4,5,6,7] and crank ergometer for upper limb work [8,9,10]. When working on a cycle or crank ergometer, the maximum power output achieved depends mainly on the magnitude of the braking force [11]. There is an obvious relationship between the amount of breaking force and the speed of rotation; the higher the applied breaking force, the lower the rotational speed. For isolated muscles or monoarticular loading, there is a hyperbolic relationship between force and speed, but for work on a cycle or crank ergometer, the force–speed relationship can be linearized in a limited range (approx. 100 to 200 rpm^−1^) [4,12]. The relationship between force and velocity can be expressed as follows: v = v_0_ (1 − F/F_0_) and F = F_0_ (1 − v/v_0_), where v_0_ and F_0_ are constants that correspond to the intersection with the velocity axis and the force axis. The parameter v_0_ corresponds to the theoretical maximum velocity that would be achieved at zero breaking force, while F_0_ represents the breaking force corresponding to zero velocity. Maximum anaerobic power (Pmax) can be achieved at approximately 50% F_0_ and 50% v_0_; therefore, it can be determined according to the relationship Pmax [W] = (0.5 × v_0_ [rpm^−1^] × 0.5 F_0_ [N]/9.81 [m·s^−2^]. The principle of the force–velocity test (F-v test) on a cycle or crank ergometer is to determine the individual rotational frequencies that correspond to different braking forces. The test consists of a series of short sprints on a cycle or crank ergometer lasting about 7 s and at a range of breaking forces, e.g., 30, 40, 50, 60 and 70 N, which should allow for the determination of an individual force–velocity relationship. Recovery breaks between the individual sprints always last 5 min. By extrapolation of the linearized relation of force and velocity, the intersections with the x and y axes can be determined, respectively. Subsequently, with the help of the parameters v_0_ and F_0_, the maximum anaerobic power can be determined [4,5].

The force–velocity test is widely used to evaluate prerequisites for anaerobic performance in athletes of different orientations and disciplines [13,14,15,16], including evaluations of disabled athletes [17,18]. Some authors [13,15] state, as one of the advantages of the force–velocity test, that, unlike most widely used tests of anaerobic power, the force–velocity test does not require maximal effort and does not induce high metabolic acidosis, since maximal anaerobic power is not determined directly by the single anaerobic test but is determined indirectly using a series of short-term non-exhaustive loads. Another advantage of the force–velocity test is that, in addition to the maximum power output, it also provides information on its speed and force components and allows for a comprehensive assessment and evaluation of all three parameters during training preparation [19] and in injury prevention and rehabilitation [20].

In general, it can be assumed that the differences in the force–velocity characteristics of the upper and lower limbs will be mainly due to different amounts of muscle mass and the composition of the skeletal muscle [3,11]. The speed of contraction depends mainly on the representation of fast-twitch muscle fibers (type II), and the level of maximum anaerobic power also depends on the degree of representation of fast-twitch muscle fibers and the activity of enzymes of anaerobic metabolism [11]. The influence of physical exercises on a significant number of physiological reactions in the body has not been sufficiently studied and analyzed at the level of cells, tissues and organs. The immediate physiological response to moderate physical exertion and/or intense physical activity depends on the type, duration, intensity and type of exercise, as well as the individual levels of training status of the participants and their individual hemorheological characteristics [21]. Force–velocity effects are probably combined with the effects of performed physical activity on the biomechanical and fluid properties of blood and blood cells, e.g., erythrocyte deformability and aggregation, changes in blood flow (through vasodilatation and change in overall blood viscosity), changes in the endothelial cells of the vascular walls, changes in blood pressure and erythrocyte release of ATP after performed anaerobic force–velocity tests of the upper and lower limbs [22].

The force–velocity characteristics of limbs were mainly studied in isolation for upper or lower limbs using a force–velocity test regarding sport orientation [11,12,13,14], but data comparing the speed and force characteristics during the work of upper and lower limbs in the same person are still very limited. Jemni et al. [14], for example, found the maximum anaerobic power in the force–velocity test of the upper limbs at a level of 65 to 67% of the maximum anaerobic power of the lower limbs in elite male gymnasts. In a group of top athletes—javelin throwers—Bouhel et al. [20] described the maximum anaerobic power, F_0_ and v_0_ in the force–velocity test of the upper limbs at a level of 60%, 67% and 109% of the values found in the force–velocity test of the lower limbs, respectively. In a group of male kickboxers, Nikolaidis et al. [15] found the maximum anaerobic power, F_0_ and v_0_ in the force–velocity test of the upper limbs at a level of 46%, 57% and 83% of the values found in the force–velocity test of the lower limbs, respectively. In contrast to groups of athletic populations of various sport orientations, there is not sufficient evidence on force–velocity data in the normal population without special sport training, especially in females. The aim of the study was, therefore, to compare the force–velocity characteristics of the upper and lower limbs in the normal population in both males and females by comparing data from the force–velocity test of the lower and upper limbs. The work hypothesis assumed that the power and force characteristics found in the upper limb test would be significantly lower than in the lower limb test (corresponding at about 70%), while the speed characteristics found in the upper and lower limb force–velocity tests would be comparable, with a difference less than 10%. The secondary hypothesis was that power outputs and F_0_ parameters in the upper and lower limb tests would be correlated both in men and women.

## 2. Materials and Methods

The research group consisted of 32 males (23.2 ± 2.8 years old; 75.0 ± 9.4 kg; 178 +/8 cm; body fat of 11.7 ± 2.8%; fat-free mass of 66.1 ± 7.2 kg) and 22 female healthy volunteers (22.7 ± 1.4 years old; 62.0 ± 6.1 kg; 168 +/6 cm; body fat of 18.9 ± 3.2%; fat-free mass of 50.6 ± 5.2 kg), who were students of Physiotherapy, Sport Management and Physical Education. None of the subjects were involved in regular exercise training. The study was approved by the ethics committee of the Faculty of Physical Education and Sport, Charles University (reference number 126/2015, approved 17 September 2015), and all the measurements were performed according to the ethical standards of the Helsinki Declaration. The subjects were fully informed in advance regarding the objectives of the study, as well as that the study methods involved no risks, and written consent was obtained from each subject for participation in the study.

All volunteers visited the laboratory twice. During the first session, there was an anthropometric examination and familiarization with the force–velocity test. Subsequently, after a suitable and adequate warm-up, the first test took place. The type of test was randomly chosen, either as a test of the lower or upper limbs. Subsequently, with an interval of several days, a maximum of one week, the second test took place. If the force–velocity test of the upper limbs was carried out in the first examination, the second test was focused on the work of the lower limb, and vice versa. All the tests were performed at the same time of day to minimize the effects of circadian rhythm [21] and with similar environmental conditions for all participants (the mean temperature and humidity were 21 ± 1 °C and 45 ± 4%, respectively).

Body height (cm) was measured by a digital stadiometer Seca 242 (Vogel & Halke, Hamburg, Germany) to the nearest 0.1 cm. Body weight was measured on a digital scale to the nearest 0.1 kg, and body mass index (BMI, in kg/m^2^) was calculated. The percentage of body fat and the amount of fat-free mass were determined on the basis of measurements of 10 skinfolds [23].

The anaerobic force–velocity test of the upper limbs was completed on a modified mechanically braked Monark-type crank ergometer, and the force–velocity test of the lower limbs was completed on a Monark 824E mechanical cycle ergometer. Both ergometers were calibrated at a wide range of speeds and loads before the actual study. After the initial basic warm-up, i.e., 10-minute-long dynamic stretching exercises, either for legs or arms [24], individual adjustments were made on both ergometers. On the ergometer for the test of the upper limbs, individual adjustments were made for the seat position and crank length (elbows were slightly bent when the arm was outstretched), and the axis of rotation of the ergometer was adjusted to ensure alignment between the ergometer crankshaft and the center of the participant’s glenohumeral joint [25]. In addition, both feet of the examined person were fixed to ensure stability and optimal conditions for working on the ergometer. In the case of the cycle ergometer, the height and the front-back adjustment of the saddle, the position of the handlebars and the fixation of the feet on the pedals were individually adjusted using clips and straps. This was followed by another phase of warm-up when the participants performed 5 min of cycling (80 W and 50 W for men and women, respectively) before the leg test or arm cranking (50 W and 20 W for men and women, respectively) for arm tests, with two accelerations lasting 3 s at the end of the third and fifth minutes. After 5 min of passive recovery, the participants performed the force–velocity tests, which consisted of five repetitive short maximal sprints against increasing braking forces lasting 7 s and interspersed breaks lasting 5 min. The anaerobic force–velocity test of the work of the upper limbs was completed using ascending braking loads (2, 3, 4, 5 and 6 kg, or 19.6, 29.4, 39.2, 49.0 and 58.8 N), and the test of the lower limbs was completed similarly using five ascending breaking loads (3, 4, 5, 6 and 7 kg, or 29.4, 39.2, 49.0, 58.8 and 68.6 N). For women, braking forces were reduced by 1 kg or 9.81 N, so the test of the upper limbs started at 9.81 N, and the test of lower limbs started at 19.6 N. If the subject was not able to reach a peak velocity higher than 100 rpm, the test was terminated prematurely, excluding higher levels of load. During the test of the lower limbs, the volunteers were required to sit on the saddle of the ergometer. During the entire test, the subjects were motivated and supported to perform sprints as quickly as possible. Specialized software registered online performance in each revolution, and the peak speed achieved in each sprint was subsequently used to construct a linear force velocity relationship as v = v_0_ (1 − F/F_0_) and F = F_0_ (1 − v/v_0_) and determine the theoretical maximum force at zero speed F_0_ and the theoretical maximum speed v_0_ corresponding to the zero braking force. Maximum anaerobic power was determined as the product of optimal force 0.5 F_0_ and optimal speed 0.5 v_0_, i.e., Pmax = 0.25 · F_0_ · v_0_ [4,5].

Data analysis: Basic descriptive statistics (mean, standard deviation) were computed for all variables, which were subsequently tested for normality using Shapiro–Wilk tests. Differences between the force–velocity characteristics were evaluated using an index of effect size—ES (Cohen’s “d”). The effect size (ES) was assessed as follows: ES < 0.20 as a small effect; ES from 0.2 to 0.8 as a medium effect; and ES > 0.80 as a large effect. The relationship between the variables obtained was evaluated using Pearson’s correlation coefficient. Statistical analyses were performed using Microsoft Excel (210) and SPSS version 22 (SPSS Inc., Chicago, IL, USA).

## 3. Results

The anthropometric characteristics of the participants are shown in Table 1. The force–velocity characteristics of the male participants’ lower and upper limbs are presented in Table 2. The lower and upper limbs differed with regard to Pmax, and the relative values of Pmax (rPmax), v_0_, F_0_, v_0_/F_0_ and effect size showed a medium to large effect. While the parameters Pmax, rPmax, v_0_ and F_0_ were higher in the lower limb test, on the other hand, the parameter v_0_/F_0_ reached higher values in the upper limb test than in the lower limb exercise test. Table 3 shows the force–velocity characteristics of the female participants’ lower and upper limbs. Similar to the group of men, the parameters Pmax, rPmax, v_0_ and F_0_ were higher in the test of the lower limbs than in the test of the upper limbs, and the parameter v_0_/F_0_ reached higher values in the upper limb test than in the lower limb exercise test. Table 4 documents the ratio between the upper and lower limb force–velocity test results, where the upper limb test results are expressed as a percentage of the lower limb test results. The table also compares the percentage results achieved by men and women and shows that in the force–velocity test, men achieve significantly higher values of absolute and relative power output (a large effect), and the values of the parameters v_0_ and F_0_ are somewhat higher in men than in women (a medium effect). The parameter v_0_/F_0_, however, was not different in males and females.

A correlation analysis of men showed a significant correlation between the relative maximal power output (rPmax) and parameter F_0_ in the lower limb force–velocity test (r = 0.372; *p* < 0.05) but not in the upper limb force–velocity test (r = 0.323; n.s.). There were no significant correlations between the relative maximal power output (rPmax) and parameter v_0_, neither in the force–velocity test for the lower limbs nor in the test for the upper limbs (r = −0.190; n.s.; r = −0.166; n.s., respectively). Comparing the tests of the lower and upper limbs in men, no significant correlation was found between the relative maximal power outputs (rPmax) (r = 0.232; n.s.) and parameter v_0_ (r = 0.048; n.s.), but the F_0_ parameters in both tests were correlated with each other (r = 0.444; *p* < 0.05). A correlation analysis of women showed stronger correlations than in men. Significant correlation was found between the relative maximal power output (rPmax) and parameter F_0_ both in the lower limb force–velocity test (r = 0.703; *p* < 0.01) and in the upper limb force–velocity test (r = 0.891; *p* < 0.01). The relative maximal power output (rPmax) and parameter v_0_ were correlated both in the force–velocity test for the lower limbs and in the test for the upper limbs (r = −0.519; *p* < 0.05; r = −0.503; *p* < 0.05, respectively). Comparing the tests of lower and upper limbs in women, no significant correlation was found between the relative maximal power outputs (rPmax) (r = 0.421; n.s.) and parameter v_0_ (r = 0.387; n.s.), but the F_0_ parameters in both tests were correlated with each other (r = 0.444; *p* < 0.05).

## 4. Discussion

Our study is one of the few studies [26] comparing the characteristics of the force–velocity test in men and women and, at the same time, comparing the data of the lower and upper limb tests. There are a number of studies in the sport science literature, usually in the male sports population of various orientations, where it is usually tests of the lower, and more rarely upper, limbs. Data on force–velocity characteristics in women are relatively rare compared to male athletes. In only a few studies, women were tested, either only with the force–velocity test of the lower or only the upper limbs. Our study demonstrated that Pmax, rPmax, F_0_, v_0_ and v_0_/F_0_ differed significantly between the upper and lower limbs in both males and females. Pmax, rPmax, F_0_ and v_0_ were higher in the lower extremities, whereas v_0_/F_0_ was higher in the test of the upper extremities. This finding is in agreement with the data of other studies, such as [15], who, based on similar results, formulated that the upper limbs show the so-called “faster” profile and the lower limbs, on the contrary, show a “stronger” profile.

The main results of the study confirmed the work hypothesis that the power and force characteristics found in the upper limb test would be significantly lower than in the lower limb test (roughly corresponding to about 70%), while the speed characteristics found in the upper and lower limb force–velocity tests were comparable and differed less than 10%. The maximum anaerobic power (Pmax) in the upper limb test was lower than in the lower limb test and corresponded to 78.2% and 67.5% of the lower values of Pmax in males and females, respectively. The respective values of the parameter F_0_ in the upper limb test were lower than in the lower limb test and corresponded to 80.1% and 74.5% of the lower test F_0_ values in males and females, respectively. Our hypothesis also assumed that the speed characteristics v_0_ found in the upper and lower limb force–velocity tests would be comparable. The result of the study showed that v_0_ values were only slightly lower in the test of the upper limbs than in the test of the lower limbs, and their comparison showed only a medium effect. The secondary hypothesis was that power outputs and F_0_ parameters in the upper and lower limb tests would be correlated in both men and women. This assumption was confirmed for both men and women only in the case of the F_0_ parameter but not in the power outputs in the upper and lower limb tests.

The values of the theoretical maximum force F_0_ for the upper and lower limbs in males of 173.1 and 218.9 N in our study were somewhat higher than those found in recreational athletes, 112 N and 140 N in the studies of Vandewalle et al. [5,9] or in non-competitive male boxers, 102 N and 168 N [27], but were close to the values of 133 N and 239 N found in kickboxers [15] for the upper limbs and lower limbs, respectively. Similar and/or higher values of F_0_ for upper limbs in our study were found in senior kayak paddlers (182 N) and/or in canoe paddlers (155 N) [17]. The value of F_0_ for the upper limbs in females, 110.6 N, was comparable to the one found for the upper limbs in female kayak paddlers, 107 N [17]. The value of F_0_ for the lower limbs, 148.3 N, however, was higher than the value of F_0_ for the upper limbs, 111 N, reported in females aged 21 to 27 years [4]. The value v_0_, i.e., the theoretical maximum speed at zero breaking force for upper and lower limbs in males, attained 209.3 and 214.4 rpm, which is comparable to the results for upper limbs (228 rpm in male students [28]; 254 rpm in young swimmers [9]) as well as for lower limbs (211 rpm in male students [28]; 216 rpm in endurance athletes [29]; and/or 228 rpm in recreationally active men [4]). The value of v_0_ in females for upper limbs, 185.9 rpm, was slightly lower than that reported in female students (202 rpm [26]) or female kayak paddlers (218 rpm [17]). The value of v_0_ in females for lower limbs, 186.9 rpm, was also slightly lower than the result found for lower limbs in females aged 21 to 27 years—211 rpm [4].

The results of Pmax for the upper extremities (845.3 ± 111.4 W) in males were comparable to the reference data (790 W [27]; 718 W [9]; 783 W [24]; and 960 W [29]). The corresponding values for the lower extremities (1097.6 ± 155.6 W) were similar to other reported data (1244 W [27]; 1180 W [30]; 1165 W [15]; 1111 W [25]; and 1080 W [28]). In females, the values of Pmax for the upper limbs, 518.6 ± 87.2 W, were slightly lower than those reported in female kayak paddlers (583 W [17]) but higher than in female students (380 W [26]). The values of Pmax in females for lower limbs, 763.4 ± 81.9 W, were somewhat higher than those reported for female students (594 W [8]; 662 W [26]) or female kayak paddlers (583 W [17]). The relative value of Pmax (rPmax) for upper limbs measured in men with the force–velocity test was 11.4 ± 1.7 W·kg^−1^, which is comparable with the values found in students (10.7 W·kg^−1^ [28]; 12.3 W·kg^−1^ [31]; 10.2 W·kg^−1^ [25]) or swimmers (10.1 W·kg^−1^ [9]). The relative value of Pmax (rPmax) for lower limbs, 14.8 ± 2.3 W·kg^−1^, was comparable to previous reported data (14.7 W·kg^−1^ [13]; 14.5 W [25]; 16.4 W·kg^−1^ [28]; 15.3 W·kg^−1^ [15]; 13.7 W·kg^−1^ [29]; or 12.4 W·kg^−1^ [4]). The relative value of Pmax (rPmax) for upper limbs measured in women was 8.0 ± 1.4 W·kg^−1^, which is comparable with the values reported in female kayak paddlers (9.5 W·kg^−1^ [17]) or female students (6.3 W·kg^−1^ [26]). The relative value of Pmax (rPmax) for lower limbs found in women was 11.8 ± 2.2 W·kg^−1^, which is comparable to the values reported in female students (10.8 W·kg^−1^ [26]). The upper-to-lower extremities ratio in males with regard to Pmax, 78.2 ± 14.3%, was higher than in male kickboxers (46.4% [15]) or recreational boxers (49% [27]) but comparable to those found in physical education students (70.5% [26]; 65.1% [28]). The upper-to-lower extremities ratio in females, 67.6 ± 12.6%, was somewhat higher than the one reported in female students (57.5% [26]). The differences between force–velocity characteristics of upper and lower limbs could be explained primarily due to muscle mass and distribution of muscle fibers, but also peripheral hemodynamic processes may play an important role [22]. It has been proposed that muscle mass in the upper limbs corresponds to approximately 22% and 17% of muscle mass in males and females, respectively [32]. Muscle strength, or force-generating capacity, is closely related to muscle mass and muscle cross-sectional areas [18,28]. While the differences in force might be attributed to variations in muscle mass, the difference in velocity could be due to variations in fast-twitch muscle fiber distribution [3,18]. With regard to the associations between force–velocity characteristics of upper and lower limbs, we have found significant correlations for absolute or relative values of maximum anaerobic power and parameter F_0_, but not for parameter v_0_. This finding is comparable to the results of similar studies, which indicated a closer relationship between upper and lower limb power and strength force than for the velocity parameter v_0_ [15,27,28].

### Strengths and Limitations of the Study

The study brings data on the force–velocity test of legs and arms in healthy and moderately active young adults, men and women, without specialized training. These data may be used as reference data for male and female athletes to evaluate whether their functional status during training is comparable to or higher than in the average population. The data could also be used in rehabilitation programs for athletes, e.g., in the assessment of possible strength deficits in various areas of the force–velocity characteristic spectrum of the muscles of the upper or lower limbs. The main limitation of the present study is the number of volunteers (men and women), which was not large enough for the achieved results to be used as reliable reference data. In further studies, it would be desirable to investigate population groups of different ages and higher or lower levels of physical activity. Although the reliability of the characteristics of the force–velocity test in cycling and cranking seems to be high [26], an ideally performed force–velocity test requires some technical experience in such a type of short, intense exercise; therefore, it would be appropriate to examine subjects not only once, i.e. cross-sectionally, but for example twice or repeatedly.

## 5. Conclusions

The studies on the relationship between force and speed of muscle contraction and muscle work in men and women may expand the knowledge of muscle physiology. A comparison of the force–velocity characteristics of the muscles of the upper and lower limbs in recreationally physically active men and women using the arm cranking and leg cycling force–velocity test showed that the force characteristics of the test F_0_ and the derived maximum anaerobic power in the upper limb test reached 80% and 78% in men and 74% and 68% in women of the values determined in the test of lower limbs for the F_0_ and Pmax, respectively. In contrast to power and strength indices, the speed characteristics of the force–velocity test of the upper and lower limbs were not substantially different. The results of the study can serve as reference data for the force–velocity characteristics of the upper and lower limbs of male and female athletes. The results can be used both in training and rehabilitation programs, where the starting point is the objectification of possible strength deficits in various areas of the force–velocity characteristic spectrum of the muscles of the upper and lower limbs.

## Figures and Tables

**Table 1 jfmk-08-00151-t001:** Characteristics of the participants (means and SDs).

	Males (*n* = 32)	Females (*n* = 22)
Age [years]	23.2 (2.8)	22.7 (1.4)
Weight [kg]	75.0 (9.4)	62.6 (6.1)
Height [cm]	178.2 (8.3)	158.0 (6.0)
BMI [kg/m^2^]	23.5 (1.5)	22.2 (1.7)
Body fat [%]	11.7 (2.8)	18.9 (3.2)
Fat-free mass [kg]	66.1 (7.2)	50.6 (5.2)

**Table 2 jfmk-08-00151-t002:** Force–velocity characteristics in males (*n* = 32; means and SDs).

	Lower Limbs	Upper Limbs	d-Value	Effect Size
Pmax [W]	1097.6 (155.6)	845.3 (111.4)	1.83	large
rPmax [W/kg]	14.8 (2.3)	11.4 (1.7)	1.71	large
v_0_ [rpm]	214.4 (14.1)	209.3 (28.0)	0.23	medium
F_0_ [N]	218.9 (41.1)	173.1 (40,2)	1.11	large
v_0_/F_0_ [rpm/N]	1.01 (0.19)	1.30 (0.35)	0.89	large

**Table 3 jfmk-08-00151-t003:** Force–velocity characteristics in females (*n* = 22; means and SDs).

	Lower Limbs	Upper Limbs	d-Value	Effect Size
Pmax [W]	763.4 (81.9)	518.6 (87.2)	2.18	large
rPmax [W/kg]	11.8 (2.2)	8.0 (1.4)	2.01	large
v_0_ [rpm]	195.1 (14.4)	185.9 (13.7)	0.36	medium
F_0_ [N]	148.3 (30.5)	110.6 (22,8)	1.11	large
v_0_/F_0_ [rpm/N]	1.39 (0.39)	1.78 (0.41)	0.89	large

**Table 4 jfmk-08-00151-t004:** Upper-to-lower limb force–velocity characteristics ratio (*n* = 22; means and SDs).

	Lower Limbs	Upper Limbs	d-Value	Effect Size
Pmax [W]	78.2 (14.3)	67.6 (12.6)	0.90	large
rPmax [W/kg]	78.2 (14.3)	67.5 (12.5)	0.92	large
v_0_ [rpm]	97.9 (14.0)	92.3 (9.6)	0.46	medium
F_0_ [N]	80.1 (17.3)	74.5 (6.5)	0.39	medium
v_0_/F_0_ [rpm/N]	129.1 (29.0)	127.40 (26.4)	0.06	small

## Data Availability

The input data are with the authors and can be provided in anonymised form to anyone who shows an interest in them.

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
