# Peer review of "Force–Velocity Relationship in Cycling and Arm Cranking: A Comparison of Men and Women"

_jfmk, 2023, doi:10.3390/jfmk8040151_

Round 1
Reviewer 1 Report
Comments and Suggestions for Authors
The paper is correctly written in a methodological way, the document aims to: Study was aimed at comparison of the force-velocity relationship during cycling and arm 11 cranking in males and females.
The research authors are asked to solve the following questions:
1) Specify the research objective in the last paragraph of the introductory section.
2) It is necessary to specify the study importance, in terms of improving the theory and methodology of cycling sports training. It is important that the research authors ask themselves: What will the research contribute? Since a comparison between genders is not useful if it does not have a sense of future improvement in some aspect of the preparation.
3) In the last paragraph of the discussion, the research limitations are specified, but the strengths should be specified summarily.
4) To optimize reading ability, it is recommended to create a subsection named “Strengths and limitations of the research.”
5) The conclusions are not justifying (Line: 288-289). The research justification must be in the introduction section (See question 2), going deeper, if necessary, in the discussion section, confirming with the criteria of other authors.
Comments on the Quality of English LanguageConsult with an English language specialist
Author Response
Reviewer 1
The authors thank the reviewer for stimulating and valuable comments to improve the quality of the submitted manuscript.
The research authors are asked to solve the following questions:
- Specify the research objective in the last paragraph of the introductory section.
Introductory section was rewritten and research objective was better specified.
- It is necessary to specify the study importance, in terms of improving the theory and methodology of cycling sports training. It is important that the research authors ask themselves: What will the research contribute? Since a comparison between genders is not useful if it does not have a sense of future improvement in some aspect of the preparation.
Contribution of the study for sports training and practical aspects of the study were corrected and specified.
3) In the last paragraph of the discussion, the research limitations are specified, but the strengths should be specified summarily.
4) To optimize reading ability, it is recommended to create a subsection named “Strengths and limitations of the research.”
New paragraph named “Strengths and limitations of the research” was added and strengths and limitations were presented in detail.
5) The conclusions are not justifying (Line: 288-289). The research justification must be in the introduction section (See question 2), going deeper, if necessary, in the discussion section, confirming with the criteria of other authors.
The conclusions were rewritten and corrected.
Reviewer 2 Report
Comments and Suggestions for Authors
Title Force-velocity relationship in cycling and arm cranking: A comparison of men and women
The manuscript topic is important and very interesting.
Оbligatory corrections:
1. The research group consisted of 32 males and 22 female healthy volunteers. Whole group characteristics – anthropometric, mass etc. data must be present in one big table for clearness.
2. The References can be enriched with the same from the last 5 years - only 1 reference from 30 is from the last five years.
3. The Abstract and Conclusions must be improved.
Improvement. Based on the proposed corrections and others, these two paragraphs should be rewritten clearly underline the new results and conclusions presented from the authors. In the Abstract must be included the main conclusions and the authors must underline the own approach contributions and the basic benefits from presented results. The unwritten rule is that most readers only look at these paragraphs – abstract and conclusions.
Recommended corrections for manuscript quality increase:
1. In the rows 65-70 the authors wrote “In general, it can be assumed that the differences in the force-velocity characteristics of the upper and lower limbs will be mainly due to the different amount of muscle mass and the composition of the skeletal muscle [18]. The speed of contraction depends mainly on the representation of fast twitch muscle fibers (type II) and the level of maximum anaerobic power also depends on the degree of representation of fast muscle fibers and the activity of enzymes of anaerobic metabolism [18,19].”
It is a well-known fact that the influence of physical exercises on a significant number
of physiological reactions of the body has not been sufficiently studied and analyzed at
the level of cells, tissues and organs. The immediate physiological response to moderate
physical exertion and/or intense physical activity depends on the type, duration, intensity
and cycling of physical exercise, as well as on the individual levels of training status of the
participants and individual hemorheological characteristics [Ivanov, I. Hemorheological Alterations and Physical Activity. Appl. Sci. 2022, 12, 10374. https://doi.org/10.3390/app122010374].
The evaluated force-velosity effects are probably combined with the effects of done physical activity on biomechanical and fluid properties of blood and blood cells:
- Erythrocyte deformability and aggregation;
- Changes in blood flow (through vasodilatation and change in overall blood viscosity);
- Changes in the endothelial cells of the vascular walls;
- Changes in blood pressure;
- Erythrocytes release of ATP after done anaerobic force-velocity test of upper and lower limbs.
2. Rows 120-122. Authors wrote “After the initial basic warm-up (dynamic stretching exercises), the individual adjustments were made on both ergometers.” Here the initial basic personal warm-up with dynamic stretching exercises must be clearly described. It is very important to what extent each participant has performed the warm-up exercises such as repetitions, intensity and duration. Hence the force-velocity results will probably be different.
In the present paper for in depth analysis of the presented force-velocity data can include the short note in the direction of hemorheological personal status influence on the obtained results [Ivanov, I. Hemorheological Alterations and Physical Activity. Appl. Sci. 2022, 12, 10374. https://doi.org/10.3390/app122010374]. It must be very interesting if in the future study the authors combine the force-velocity profile with hemorheological status (blood characteristics, mainly whole blood viscosity).
The proposed experimental model in my opinion has high potential for implementation in practice for muscle and joint pathology evaluation.
I hope that the proposed corrections will increase the quality of the manuscript and possibly its citability.
Author Response
Reviewer 2
The authors thank the reviewer for stimulating and valuable comments to improve the quality of the submitted manuscript.
Оbligatory corrections:
- The research group consisted of 32 males and 22 female healthy volunteers. Whole group characteristics – anthropometric, mass etc. data must be present in one big table for clearness.
New table on whole characteristics on anthropometric data was added for clearness (Table 1).
- The References can be enriched with the same from the last 5 years - only 1 reference from 30 is from the last five years.
Literary references have been critically revised and newer literary sources from the last five years have been added.
- The Abstract and Conclusions must be improved.
Improvement. Based on the proposed corrections and others, these two paragraphs should be rewritten clearly underline the new results and conclusions presented from the authors. In the Abstract must be included the main conclusions and the authors must underline the own approach contributions and the basic benefits from presented results. The unwritten rule is that most readers only look at these paragraphs – abstract and conclusions.
The abstract and the conclusions of the work have been modified in such a way as to underline the new knowledge and benefits of the presented study, so that the reader can be more accurately informed about the conclusions, benefits and significance of the presented study.
Recommended corrections for manuscript quality increase:
- In the rows 65-70 the authors wrote “In general, it can be assumed that the differences in the force-velocity characteristics of the upper and lower limbs will be mainly due to the different amount of muscle mass and the composition of the skeletal muscle [18]. The speed of contraction depends mainly on the representation of fast twitch muscle fibers (type II) and the level of maximum anaerobic power also depends on the degree of representation of fast muscle fibers and the activity of enzymes of anaerobic metabolism [18,19].”
It is a well-known fact that the influence of physical exercises on a significant number
of physiological reactions of the body has not been sufficiently studied and analyzed at
the level of cells, tissues and organs. The immediate physiological response to moderate
physical exertion and/or intense physical activity depends on the type, duration, intensity
and cycling of physical exercise, as well as on the individual levels of training status of the
participants and individual hemorheological characteristics [Ivanov, I. Hemorheological Alterations and Physical Activity. Appl. Sci. 2022, 12, 10374. https://doi.org/10.3390/app122010374].
The evaluated force-velosity effects are probably combined with the effects of done physical activity on biomechanical and fluid properties of blood and blood cells:
- Erythrocyte deformability and aggregation;
- Changes in blood flow (through vasodilatation and change in overall blood viscosity);
- Changes in the endothelial cells of the vascular walls;
- Changes in blood pressure;
- Erythrocytes release of ATP after done anaerobic force-velocity test of upper and lower limbs.
The authors thank the reviewer for the appropriate addition to the discussion of the mechanisms responsible for the differences in speed-force characteristics by hematological and hemorheological parameters. Changes in hematological and hemorheological parameters in combination with biomechanical mechanisms make it possible to better approximate and explain the differences found in the speed-force characteristics in the exercise models in the presented study.
- Rows 120-122. Authors wrote “After the initial basic warm-up (dynamic stretching exercises), the individual adjustments were made on both ergometers.” Here the initial basic personal warm-upwith dynamic stretching exercises must be clearly described. It is very important to what extent each participant has performed the warm-up exercises such as repetitions, intensity and duration. Hence the force-velocity results will probably be different.
Information on dynamic stretching was corrected and presented more in detail.
In the present paper for in depth analysis of the presented force-velocity data can include the short note in the direction of hemorheological personal status influence on the obtained results [Ivanov, I. Hemorheological Alterations and Physical Activity. Appl. Sci. 2022, 12, 10374. https://doi.org/10.3390/app122010374]. It must be very interesting if in the future study the authors combine the force-velocity profile with hemorheological status (blood characteristics, mainly whole blood viscosity).
The proposed experimental model in my opinion has high potential for implementation in practice for muscle and joint pathology evaluation.
I hope that the proposed corrections will increase the quality of the manuscript and possibly its citability.
Reviewer 3 Report
Comments and Suggestions for Authors
Overall, the work is well-structured and easy to read. However, it seems to me that it doesn't provide a detailed justification for the relevance of the study or its effective contribution to increasing knowledge in this area. The authors state that: 'In general, it can be assumed that the differences in the force-velocity characteristics of the upper and lower limbs will be mainly due to the different amount of muscle mass and the composition of the skeletal muscle.' This fact is self-evident. They do not provide further justifications for this fact. It would be very opportune to explore these justifications more thoroughly.
"The working hypothesis assumed that the power and force characteristics found in the upper limb test would be significantly lower than in the lower limb test (corresponding to about 70%), while the speed characteristics found in the upper and lower limb force-velocity test would be more or less comparable." This hypothesis is self-evident. What other result could we expect? What does 'more or less comparable' mean? The authors also do not justify why it is important to conduct this study in non-athlete individuals. The work would be much more interesting if it were applicable in a specific context. The results are as expected: more strength in the lower limbs; better results in men. In my opinion, for the study to have higher quality, it would need to attempt to associate the results with other variables. This could be easier to accomplish if the sample consisted of athletes in a specific sports discipline.
Author Response
Reviewer 3
The authors thank the reviewer for stimulating and valuable comments to improve the quality of the submitted manuscript.
Overall, the work is well-structured and easy to read. However, it seems to me that it doesn't provide a detailed justification for the relevance of the study or its effective contribution to increasing knowledge in this area. The authors state that: 'In general, it can be assumed that the differences in the force-velocity characteristics of the upper and lower limbs will be mainly due to the different amount of muscle mass and the composition of the skeletal muscle.' This fact is self-evident. They do not provide further justifications for this fact. It would be very opportune to explore these justifications more thoroughly.
Introductory section was rewritten and research objectives were better specified, including contribution of the study for sports training and testing of athletes.
"The working hypothesis assumed that the power and force characteristics found in the upper limb test would be significantly lower than in the lower limb test (corresponding to about 70%), while the speed characteristics found in the upper and lower limb force-velocity test would be more or less comparable." This hypothesis is self-evident. What other result could we expect? What does 'more or less comparable' mean? The authors also do not justify why it is important to conduct this study in non-athlete individuals. The work would be much more interesting if it were applicable in a specific context. The results are as expected: more strength in the lower limbs; better results in men. In my opinion, for the study to have higher quality, it would need to attempt to associate the results with other variables. This could be easier to accomplish if the sample consisted of athletes in a specific sports discipline.
Working hypothesis was rewritten and better specified, secondary hypothesis was added.
Importance for conducting study in non-athlete individuals was added. The text also brings an information on the benefits of “reference” data from the study for sport training an rehabilitation in athletes.
Round 2
Reviewer 1 Report
Comments and Suggestions for Authors
The authors have answered the questions raised
Comments on the Quality of English LanguageEvaluate with an English language specialist
Reviewer 2 Report
Comments and Suggestions for Authors
Accept in present form.
Reviewer 3 Report
Comments and Suggestions for Authors
Despite the effort to clarify some of the issues raised in review 1, the weaknesses of the study remain the same. The contribution to improving knowledge in this area remains very limited.